# Pain Behavior of People with Intellectual and Developmental Disabilities Coded with the New PAIC-15 and Validation of Its Arabic Translation

**DOI:** 10.3390/brainsci11101254

**Published:** 2021-09-22

**Authors:** Ruth Defrin, Heba Beshara, Tali Benromano, Kutaiba Hssien, Chaim G. Pick, Miriam Kunz

**Affiliations:** 1Department of Physical Therapy, School of Health Professions, Sackler Faculty of Medicine, Tel Aviv University, Tel Aviv 69978, Israel; hmasarwe@hotmail.com (H.B.); kutaibahss@gmail.com (K.H.); 2Sagol School of Neuroscience, Tel Aviv University, Tel Aviv 69978, Israel; pickc@tauex.tau.ac.il; 3Department of Anatomy, Sackler Faculty of Medicine, Tel Aviv University, Tel Aviv 69978, Israel; talibenromano@gmail.com; 4Sylvan Adams Sports Institute, Tel Aviv University, Tel Aviv 69978, Israel; 5Department of Psychology and Sociology, Medical Faculty, University of Augsburg, 86159 Augsburg, Germany; miriam.kunz@med.uni-augsburg.de

**Keywords:** keyword intellectual disability, experimental pain, PAIC-15, translation, reliability

## Abstract

Pain management necessitates assessment of pain; the gold standard being self-report. Among individuals with intellectual and developmental disabilities (IDD), self-report may be limited and therefore indirect methods for pain assessment are required. A new, internationally agreed upon and user-friendly observational tool was recently published—the Pain Assessment in Impaired Cognition (PAIC-15). The current study’s aims were: to test the use of the PAIC-15 in assessing pain among people with IDD and to translate the PAIC-15 into Arabic for dissemination among Arabic-speaking professionals. Pain behavior following experimental pressure stimuli was analyzed among 30 individuals with IDD and 15 typically developing controls (TDCs). Translation of the PAIC followed the forward–backward approach; and reliability between the two versions and between raters was calculated. Observational scores with the PAIC-15 exhibited a stimulus–response relationship with pressure stimulation. Those of the IDD group were greater than those of the TDC group. The overall agreement between the English and Arabic versions was high (ICC = 0.89); single items exhibited moderate to high agreement levels. Inter-rater reliability was high (ICC = 0.92). Both versions of the PAIC-15 are feasible and reliable tools to record pain behavior in individuals with IDD. Future studies using these tools in clinical settings are warranted.

## 1. Introduction

Individuals with intellectual and developmental disabilities (IDD)—neurodevelopmental disorders that are characterized by intellectual difficulties and limitations in various aspects of living—are at an increased risk of acute and chronic pain compared to typically developing controls (TDCs) [1,2]. This risk entails, among other factors, a possible increased sensitivity to noxious stimuli [3,4], relatively high rates of injuries and falls [5,6], and secondary consequences of the IDD etiology related to painful diagnostic procedures, medical complications, use of assistive devices, etc. [7,8,9,10,11]. Consequently, the prevalence of pain among people with IDD is relatively high as concluded from proxy reports, e.g., [12,13,14].

The limited cognitive and communicative abilities of individuals with IDD present a major obstacle in quantifying their pain, which renders pain assessment and hence pain management a significant challenge [11,15]. In order not to have to rely on self-report alone, dozens of behavioral scales have been developed over the last two decades that aim to assess pain among non-communicative individuals, e.g., [16,17,18]. Although these scales are valid and reliable, some of them may require special expertise, and other scales may not necessarily be applicable to populations with differing cognitive impairments or ages, e.g., [19,20], thus limiting a comparison between populations. Furthermore, not all the scales are applicable to both acute/experimental and chronic pain states, potentially limiting the research on pain management interventions.

A very promising scale in this respect was developed via a European-funded international initiative (COST action TD1005) that took place between the years 2011 and 2017. A group of international (16 countries) and interdisciplinary researchers empirically investigated which items from established observational pain scales allowed for a reliable and valid assessment of pain in individuals with cognitive impairment (626 individuals with cognitive impairment of various etiologies and 59 controls were evaluated). The final product of this team was an internationally agreed-upon tool for Pain Assessment in Impaired Cognition (PAIC-15) [21,22]. The PAIC-15 was introduced as a meta-tool that can be used for diverse populations of individuals with cognitive impairments [22]. It also includes atypical behaviors such as freezing, that are frequent among individuals with IDD [11] but seldom appear in other scales.

However, to date, the use of the PAIC-15 has not been thoroughly examined among individuals with IDD. Therefore, one aim of this study was to compare pain behavior of individuals with IDD to that of TDCs, quantified by the PAIC-15. Based on previous studies, e.g., [3,4,23], we hypothesized that pain behavior of the former would be increased compared to TDCs. Due to the aforementioned advantages of the PAIC-15, it is important to disseminate its use worldwide. An imperative step in this direction is to translate the original English version into as many languages as possible. Thus far, the PAIC-15 has been translated into seven languages (https://paic15.com/, accessed on 21 September 2021). Therefore, another aim was to translate the PAIC-15 into Arabic and validate the translation.

## 2. Materials and Methods

### 2.1. Participants

This study included 45 adults: 30 individuals with IDD (IDD group, age 35.3 ± 6.2 years) and 15 typically developing controls (TDCs group, 31.3 ± 7.5 years). The rationale for the 2:1 allocation ratio was the much greater inter-subject variability among individuals with IDD in terms of IDD etiology and level. Individuals with IDD were recruited from day care centers for people with IDD (belonging to two organizations for people with disabilities: Alin and Elwyn). IDD was diagnosed according to clinical assessment and standardized testing of intelligence (including the Wechsler Intelligence Scale for Children-Revised and the Wechsler Preschool and Primary Scale of Intelligence) performed by a team from the Ministry of Social Affairs and Social Services, which supervises all services related to IDD. Individuals in this group had an estimated level of mild or moderate IDD and the ability to understand their mother tongue. TDCs were students and employees of Tel Aviv University or employees of the day care center for people with IDD. Exclusion criteria for all the participants were as follows: known acute or chronic pain, bruises or injuries in the testing regions (medical information on the participants with IDD was retrieved from their medical records by their legal guardian upon request, and additional information was also obtained from the primary caregiver).

This study was conducted in accordance with the Declaration of Helsinki, and the protocol was approved by the Ethics Committee of Tel Aviv University (3012/2012), the institutional review board of the Ministry of Social Affairs and Social Services (201323-01), and by the legal guardians of the participants with IDD. Prior to entering this study, a written informed consent document was obtained from all the TDCs and from the legal guardians of all the individuals with IDD, after they had received an explanation of this study’s aims and protocols. In addition, the protocol was explained to the participants with IDD and their escorts upon their arrival at the lab, and each step of the protocol was carried out only after their oral consent was obtained.

### 2.2. Instruments

#### 2.2.1. Pressure Algometer

Pressure stimuli were delivered using a hand-held pressure algometer (Somedic Sales AB, Algometer type II, Hörby, Sweden). The algometer has a built-in pressure transducer, an electronic recording and display unit, a power supply, and a subject-activated push button connected via a cable to the instrument. It has an accuracy of ±3%, and its unit of measurement is the kilopascal (kPa). The algometer operates by exerting a constantly increasing rate of pressure that is monitored by a cursor presented on the display. The size of the tip of the algometer that is pressed against the skin was 1 cm^2^. The algometer was calibrated before each measuring day.

#### 2.2.2. PAIC-15

The behavioral response to pressure stimuli was analyzed using the PAIC-15 [22]. This tool consists of 15 items divided into three behavioral domains: five items for facial expressions, five items for body movements, and five items for vocalizations. Each item has a title and an explanation to ensure the understanding of each item’s meaning. The items are scored on a 0–3 scale for magnitude of appearance: 0 = not at all, 1 = slight degree, 2 = moderate degree, and 3 = great degree. There is an additional option of “not scorable” for each item. The sum score of all the items is the final PAIC-15 score. The higher the sum score, the higher the probability the person is in pain. These 15 items are: Frowning, Narrowing eyes, Raising upper lip, Opening mouth, Looking tense, Freezing, Guarding, Resisting care, Rubbing, Restlessness, Pain-related words, Shouting, Groaning, Mumbling, and Complaining. Note, one participant with IDD turned his head during noxious stimulation, which prevented us from scoring his face items. The sum score for this participant was calculated without these items.

## 3. Procedures

### 3.1. Stimulation and Recording Procedures

The experimental protocol was designed by the experimental pain working group of the European Cooperation in the Field of Scientific and Technical Research (COST), termed “Pain assessment in patients with impaired cognition, especially dementia” (action TD1005), of which this study’s authors are members. The aims of this international group are to raise awareness of the subject of pain among individuals with cognitive impairment and to develop a pain assessment toolkit for this population. The protocol was first tested on TDCs prior to testing individuals with IDD in order to verify the intensity of the pressure stimuli and the ability to endure them for the required duration [23]. Prior to actual testing, all the participants underwent a training session in which they were familiarized with the sensation delivered via the pressure algometer and were trained in rating pain sensation with various scales. For training, the participants received pressure stimuli in the thigh region (which was not stimulated further during testing) at the same intensities used during the actual testing. In addition, the participants were instructed how to maintain their head so that their expressions would be best captured by the camera.

After a five-minute break, the experiment began. The experimental setup is described at length in our previous study [24]. The examiner stood behind the subject in order not to interfere with videotaping and to properly administer the stimuli. Each subject received three pressure stimuli of 50, 200, and 400 kPa, applied with the pressure algometer, to the upper mid part of the trapezius muscle (halfway between the neck line and the shoulder line). These stimuli were chosen based on a preliminary experiment conducted on TDCs, which was aimed to search for one innocuous stimulus (for control), one mildly noxious stimulus, and one moderately noxious pressure sensation, respectively [23,25]. Each stimulus rose rapidly from a baseline of 0 kPa to the designated intensity and lasted seven seconds (a two-second increase from baseline and five seconds in the destination intensity). The inter-stimulus interval was four minutes, in order to avoid carry-over effects between stimuli and in order to provide sufficient time for the pain rating. In addition, the examiner moved the stimulation site by approximately 0.5 cm between stimuli.

The participants were videotaped throughout the entire protocol, and the behavioral responses were analyzed retrospectively, separately for each stimulation condition (four conditions in total (rest = no stimulation, 50, 200 and 400 kPa). At baseline, the subjects were not engaged in any specific activity; the analysis of the PAIC-15 was conducted for a random 15 s segment. During pressure stimulation, the analysis commenced immediately upon the examiner starting the stimulus, and it lasted the entire duration of stimulation. The coders for the original (English) version and the Arabic translation of the PAIC-15 worked separately. The coders for the original version had a background in neuroscience and one of them has been working for several years as a sport coach of people with IDD. Both were fluent in English (one was native English speaker). The two coders of the Arabic version had a background in health professions and one of them has been, and is currently working as a physical therapist for people with IDD. Both were native Arabic speakers. Among a subsample of 10 participants, an additional rater scored pain behavior in all four of the conditions (rest, 50, 200, and 400 kPa) with the Arabic version in order to calculate inter-rater reliability.

### 3.2. The Translation Process of the PAIC-15 from English to Arabic

In order to translate the PAIC-15 into Arabic, we followed the common forward–backward approach [26]. First, a native Arabic speaker, who is also a health professional, translated the PAIC-15 from English into Arabic. The translator consulted with two additional native Arabic speakers who were fluent in English, and they all agreed upon the most adequate wording. The Arabic version was then translated back into English by two independent people who were native speakers of both Arabic and English. The re-translated English version was compared to the original English version, and in the case of disagreement between these versions, a meeting was initiated between the members of the team in order to investigate the reason for the disagreement and to discuss solutions (e.g., agreeing on one of the options or looking for another option). This meeting also included external advisers who were fluent in Arabic and English, for further consultation. Upon agreement on all the items by each team, the Arabic version was confirmed.

In the next step, we showed the Arabic version of the PAIC-15 as well as the original English version to a group of people, comprising health professionals and Arabic teachers who were all native speakers of Arabic and fluent in English. After a close inspection of the translated version and discussions among the group, it was decided that although the translated version was adequate for local Arabic speakers, it was not suitable for worldwide dissemination, the reason being that dialects of spoken Arabic can differ between countries and even between regions within a country.

Therefore, in order to comply with as many Arabic-speaking countries as possible, it was decided to prepare another version that was adjusted to the use of the Arabic language in Arabic literature which is more uniform across countries. To achieve this purpose, the two PAIC-15 versions were analyzed by a native Arabic speaker who is an English literature professor with a degree in professional translation, who suggested substituting words for some of the items. This version was then analyzed by a second professional, an Arabic language professor who is an expert in dialects and writing and who made final adjustments. For example, the term تجميد which was the initial translation for the “freezing” item was replaced by تصلب in order to refer to people, not to objects. Furthermore, the initial translation for the item “jaw dropper” يتم اسقاط الفك was replaced by خفض الفك in order to better reflect a movement that occurs as a reaction. The back translation of this version was conducted by a professor of English literature and behavioral sciences, who is a native speaker of Arabic. The final Arabic translation was introduced to a group of health professionals (nurses, occupational therapists, and physical therapists) who were all native Arabic speakers, some of whom worked with people with IDD. Upon an open discussion of all the items of this version, they found the tool universally understood and feasible, as well as adequate. This final tool (available at https://paic15.com/www-PAIC15.com, accessed on 21 September 2021) was used to code the pain behavior of the participants following pressure stimulation.

## 4. Data Analysis

Data were analyzed with the IBM SPSS statistics software (version 25, IBM, New York, NY, USA). Whereas the sum scores of the 15 items of the PAIC-15 were considered continuous, single items were considered ordinal data. Thus, the effects of group type and stimulation condition on the sum score of the PAIC-15 as well as their interaction within each language were analyzed with a repeated-measures ANOVA and corrected post hoc comparisons. The correlation between stimulation intensities and the PAIC-15 sum scores were calculated with Pearson’s r, and a comparison between the original and translated versions of the PAIC-15 sum scores were calculated using *t*-tests. Agreement between the original and translated versions for all PAIC-15 items and for single items was calculated using the interclass correlation coefficient (ICC), and comparison between versions was conducted via the Mann–Whitney U test. Given that the PAIC-15 is aimed to measure pain behavior, a reliability assessment of the total PAIC score as well as of each of the 15 items was performed for the 400 kPa (noxious) stimulation. The inter-rater reliability of the PAIC-15 Arabic version was evaluated as well. Internal consistency was evaluated with Cronbach’s alphas. Two-tailed *p*-values are reported, and *p* < 0.05 was considered significant.

## 5. Results

### 5.1. The Study Groups

Table 1 presents the participants. The IDD group did not differ from the TDCs group in age (*t*-test: *t* = 1.81, *p* = 0.08) or sex distribution (Mann–Whitney U test: z = −0.09, *p* = 0.924). Among the IDD group, there were participants with cerebral palsy (CP), Down syndrome (DS), or an unspecified IDD (UIDD), and the majority of them had a mild impairment. Medication use among the IDD group included medication for hypothyroidism (seven participants/23.3%), psychotropic medications (6/20%), muscle relaxants (6/20%), and antiepileptics (2/6.66).

### 5.2. PAIC-15 Original Version: Comparison between the IDD and TDC Groups

Figure 1 presents the sum of the PAIC-15 scores in response to pressure stimulation for the IDD and TDC groups. A repeated-measures ANOVA revealed a significant global effect of group type (F(1,35) = 13.99, *p* < 0.001), and of condition (F(3,105) = 13.1, *p* < 0.0001) on the sum PAIC-15 scores. The interaction group type X condition was also significant (F(3,105) = 3.51, *p* < 0.018), suggesting that the increase in PAIC-15 sum scores across the stimulus intensities was not uniform in the two groups.

Post hoc tests revealed a significant group effect within every stimulation condition (*p* < 0.0001 for all four conditions): the scores for the IDD group were significantly higher in all four conditions—rest (*t* = 4.81, *p* < 0.0001), 50 kPa (*t* = 3.68, *p* < 0.001), 200 kPa (*t* = 2.99, *p* < 0.01), and 400 kPa (*t* = 3.47, *p* < 0.01) (Figure 1).

Table 2 presents the frequency of occurrence of each PAIC item in the IDD compared to the TDC groups during the 400 kPa stimulation. Overall, the IDD group was more behaviorally responsive during pain. The items “raising upper lip”, “opening mouth”, “looking tense”, and “freezing” occurred significantly more frequently among the IDD than among the TDC group. There were several items that appeared only in the IDD group, including guarding, shouting, groaning, and complaining. The rest of the items similarly occurred among both groups (Table 2).

### 5.3. PAIC-15 Original Version: Correlation between Its Scores and Stimulation Intensities within Groups

Among both the IDD and TDC groups, the PAIC-15 scores correlated moderately and significantly with pressure stimulation intensity (r = 0.55, *p* < 0.001 and r = 0.45, *p* < 0.001, respectively), suggesting a stimulus–response relation for the PAIC-15. Yet, the slopes of these stimulus–response functions were different among the groups, as can be seen in Figure 1: that of the IDD group was steeper than that of the TDC group (2.37, R2 = 0.96 vs. 0.75, R2 = 0.87), manifested also by the significant group X condition interaction.

The internal consistency of the original PAIC-15 version for the present sample was high (Cronbach’s α = 0.93).

### 5.4. PAIC-15 Arabic Version: Comparison between the IDD and TDC Groups

Figure 2 presents the sum scores of the PAIC-15 Arabic version for the IDD and TDC groups. A repeated-measures ANOVA of the scores in the Arabic version revealed a significant global effect of group type (F(1,34) = 8.6, *p* < 0.01) and of condition (F(3,102) = 17.8, *p* < 0.0001) on the PAIC-15 scores, similar to those of the English version. The interaction group type X condition showed a trend toward significance (F(3,102) = 2.75, *p* = 0.056). Similar to the results obtained with the English version, individuals with IDD had significantly higher PAIC-15 scores compared to those of TDCs at rest (*t* = 2.98, *p* < 0.01), 50 kPa (*t* = 2.63, *p* < 0.01), 200 kPa (*t* = 2.50, *p* < 0.016), and 400 kPa (*t* = 2.59, *p* < 0.022) (Figure 2).

Table 2 presents the frequency of occurrence of each PAIC item in the IDD compared to the TDC group during the 400 kPa stimulation. As found for the original PAIC version, the IDD group was more responsive during pain. The items “opening mouth” and “looking tense” were scored significantly more often among the IDD than among the TDC group. Approximately half of the items occurred only among people with IDD, and two items—shouting and mumbling—did not occur in either of the groups (Table 2).

### 5.5. PAIC-15 Arabic Version: Correlation between Its Scores and Stimulation Intensities within Groups

Among both the IDD and TDC groups, the PAIC-15 scores correlated moderately and significantly with stimulation intensity (r = 0.50, *p* < 0.001 and r = 0.61, *p* < 0.0001, respectively), suggesting a stimulus–response relation for the PAIC-15. As in the case of the scores with the original version, the slopes of these stimulus–response functions were different among the groups: That of the IDD group was steeper than that of the TDC group (1.88, R2 = 0.97 vs. 0.76, R2 = 0.81), as also manifested by the significant group X condition interaction (Figure 2).

The internal consistency of the translated version of PAIC-15 for the present sample was high (Cronbach’s α = 0.89).

### 5.6. Comparison between the Original (English) and the Translated (Arabic) PAIC-15 Versions

When comparing the scores of the English and Arabic versions within groups, the scores for the IDD group (Figure 3) via use of the Arabic version were significantly lower than those via use of the English version, for rest (*t* = 4.4, *p* < 0.01) and for 50 kPa (*t* = 2.5, *p* < 0.049). However, the PAIC-15 scores of the IDD group during stimulation were similar for the two languages—for 200 kPa (*p* = 0.132) and 400 kPa (*p* = 0.08)—even though a tendency toward lower scores with the Arabic version was observed. The scores for the TDC group with the English and Arabic versions were similar (not shown).

The overall agreement between the English and the Arabic version was high (ICC = 0.89). Table 2 presents the comparison between the versions for each PAIC-15 item separately, for 400 kPa. Agreement between the two languages as measured with the ICC ranged from moderate to high, with one item having weak agreement (looking tense). For two items—shouting and mumbling—it was impossible to calculate agreement because they appeared only when scoring was performed with the English version, albeit only in the IDD group and with low frequency (13.3%). The frequency of appearance of most of the items was similar between the languages, with three exceptions: The item “raising upper lip” appeared significantly more often when scored with the English version than with the Arabic version (*p* < 0.01), and the items “shouting” and “mumbling” appeared only when scored with the English version.

### 5.7. Inter-Rater Reliability of the Arabic PAIC-15

The overall agreement between the two raters as calculated with the ICC was very high (ICC = 0.92). The agreement between the two raters in each condition separately varied somewhat: It was high for rest (ICC = 1.0), 200 kPa (ICC = 0.98), and for 400 kPa (ICC = 0.86). However, it was weak for 50 kPa (ICC = 0.35). 

## 6. Discussion

### 6.1. The PAIC-15 as a Measure of Pain for IDD

The first aim of this study was to evaluate pain behavior among individuals with IDD using the PAIC-15. The PAIC-15 has been used to code pain behavior among older persons with dementia in both experimental and clinical settings [21,22,27]. The current study is the first time in which the PAIC-15 was systematically used for the IDD population. The PAIC-15 was suitable for the task at hand; the coders could easily understand each item owing to the explanations provided on the scale and could decide whether or not a certain item appeared while they watched the videos, and whether it was scorable or not, as the latter possibility also exists in the scale.

The analysis of the PAIC-15 scores showed that, as hypothesized, people with IDD have increased pain behavior compared to TDCs during noxious pressure stimulation but also during innocuous stimulation (50 kPa) and at rest. In other words, people with IDD are overall more active in the face and body compared to people with typical development. Aligning with these results are results from previous experimental studies in which pain behavior during experimental noxious stimulation was scored with various behavioral tools. For example, we previously analyzed the same participants using the Facial Action Coding System (FACS) and found them to exhibit more facial actions compared to the control group [23,28]. Barney et al. (2015) also reported increased pain behavior among children and adolescents (average age 14.8 years, range 8–22) with Neuronal Ceroid Lipofuscinosis as compared to their siblings [29]. The same authors also reported increased reactivity to cutaneous stimuli among children with Global Developmental Delay compared to controls [30]. Increased and/or elongated behavioral responses among children and adults with IDD have also been reported in clinical settings, for example during vaccination [31,32,33,34] and painful medical procedures [35,36,37]. Although newborns with Down syndrome were slower to express pain during such procedures, their behavioral and physiological responses persisted for a longer durations and some of the responses were enhanced compared to typically developing newborns [38].

The underlying reasons for the increased pain behavior in IDD are not fully understood. It is also not clear whether the increased pain behavior reflects an increased perception of pain among these individuals or whether it incorporates a mixture of mental experiences that include, but are not restricted to pain such as anticipation, anxiety, and apprehension. Nevertheless, several studies have reported decreased pain thresholds among individuals with IDD in the laboratory setting, which suggests increased sensitivity to noxious stimuli [3,4,39,40], possibly explaining the increased behavioral responses. Others have reported similar pain thresholds among individuals with IDD and controls [41,42], suggesting that further study is needed in order to resolve this issue. Yet, increased cortical responses to noxious stimuli in IDD that were evident in imaging studies and evoked potential studies [28,43] support the aforementioned notion that individuals with IDD may be more sensitive and/or vulnerable to pain than typically developing individuals. Not mutually exclusive explanation is that the TDCs had reduced responses compared to the IDD group because they were more self-aware of being videotaped. Nevertheless, the possible impact of self-awareness may have decreased towards the actual experiment due to the long familiarization and training process in front of the camera.

Looking at the most painful stimulus—the 400 kPa—the items that were highly frequent among the participants were three from the “facial expression” subscale (raising upper lip, opening mouth, and looking tense), and one from the “body movement” subscale (freezing). These occurred among 60–73% of the participants with IDD. The “raising upper lip” and “opening mouth” items resemble the FACS items “lips part” and “lip raiser” which have also been reported to frequently appear among individuals with IDD in experimental [24,44] and clinical settings [32,33,34,45]. Freezing has also been reported as a frequent, seemingly atypical behavior among individuals with IDD during painful insults [31,44,46], and its formal incorporation in the PAIC-15 is reinforced by the present results. The item “narrowing eyes” was also frequent among the majority of the IDD group; however, its frequency was no different from its frequency in the TDC group. Notably, the relatively low frequency of other PAIC-15 items such as “resisting care”, “rubbing”, and “shouting” was probably due to the administration of experimental pain stimuli of low to moderate intensity and to the preparation and training process the participants underwent.

### 6.2. The Arabic Translation of the PAIC-15

The translation from English into Arabic was performed via known procedures as detailed by Sousa et al. [26]. Given that we used video recordings of the pain responses, it was possible to have several observers code the same material using either the original English version or the newly translated Arabic version of the PAIC-15 scale. This allowed us to test the validation of the translation by way of four analyses. First, there was a moderate significant correlation between stimulation intensity and the sum scores of the translated PAIC-15, which supports its criterion validity. Similar correlations were also calculated for the original English version. Second, the internal consistency of the translated version was high (0.89), as was that of the original version, which suggests its high reliability as a scale. Third, the agreement between the two raters who used the translated version was very high for the noxious stimulation conditions (200 and 400 kPa), suggesting that the tool provides reliable scores for pain behavior. The agreement was low for the innocuous stimulus (50 kPa), perhaps because the tool is not meant to score such a condition. Alternatively, the 50 kPa stimulation may have evoked apprehension among some of the participants, the signs of which were inconsistently detected by the raters. Fourth, the overall agreement between the original and translated version was high, and the agreement for single items ranged from moderate to high with only one of the 15 items showing low agreement (“looking tense”, ICC = 0.39). Thus, overall, the translated version demonstrated good reliability and agreement with its original version.

Interestingly, some subtle differences were observed between the English and Arabic versions. First, the item “raising upper lip” was scored significantly more often in the IDD group when using the English version compared to the Arabic version (60 vs. 16.6%). It seems unlikely that the professional or cultural background of the coders underlie this difference although this possibility cannot be dismissed. Nevertheless, the “opening mouth” item, which reflects a somewhat related action around the mouth, exhibited no differences between the versions. Two items were scored only by the original PAIC-15: shouting and mumbling. However, these items appeared among three to four participants with IDD out of the 30 participants, and mumbling appeared among one person from the TDC group. The difference in appearance between the English and the Arabic version in these items was not significant. Thus, it appears that the use of the English and Arabic versions of the PAIC-15 resulted in an overall similar identification of items that were more frequent than others, and of increased pain behavior in the IDD than the TDC group. The tendency toward somewhat lower scores in the Arabic version for the participants with IDD may have been coincidental, may be due to differences between the observers, or it may have resulted from subtle variations in the interpretation of the items in each language. Nevertheless, the translated version resulted in similar conclusions to those of the English version with regard to pain behavior among individuals with IDD.

### 6.3. Limitations

Several limitations should be considered. First, larger groups would have allowed an evaluation of whether gender affects pain behavior as coded with the PAIC-15. Second, the current study was an experimental study; hence, the results apply to acute pain behavior. Future studies may wish to test the usability of the PAIC-15 among individuals with IDD who suffer from chronic pain or during clinical pain conditions. Third, although we intended to prepare a PAIC version that would apply to all Arabic-speaking people regardless of their country of origin (hence the translation into Arabic literature), we acknowledge that there might be some variations between countries. The comparison of PAIC-15 scores between different countries/regions may provide data that would help improve the accuracy of the translation.

### 6.4. Conclusions and Clinical Implications

Although people with IDD are exposed more frequently than others to painful conditions, they tend not to report pain, or their reports may be toned down as compared to those of typically developing individuals [47,48,49]. Thus, the use of tools to analyze indirect indices of pain such as pain behavior is imperative in order to tailor adequate pain management to these individuals. The PAIC-15 has been proven to be valid and reliable in measuring pain among people with cognitive impairment due to dementia, and the present results support its use among people with IDD as well. The use of the PAIC-15 for different populations with cognitive impairment enables the implementation of a standardized approach in the study of pain perception, in identifying etiology-related unique pain responses, and in studying pain management guidelines for this population. The tool is user friendly, does not require special professional training for the observer, and can detect pain in a dose–response manner as shown herein. Considering the advantages of the PAIC-15, its dissemination is called for. As Arabic is the official or co-official language in approximately 25 countries around the world, the translation of the PAIC-15 into Arabic is another step in the right direction, not only for people with IDD but also for people with other types of cognitive impairment. Hopefully, this scale will aid Arabic-speaking caregivers to identify pain, assess its magnitude, and provide proper treatment in order to prevent suffering among these individuals.

## Figures and Tables

**Figure 1 brainsci-11-01254-f001:**
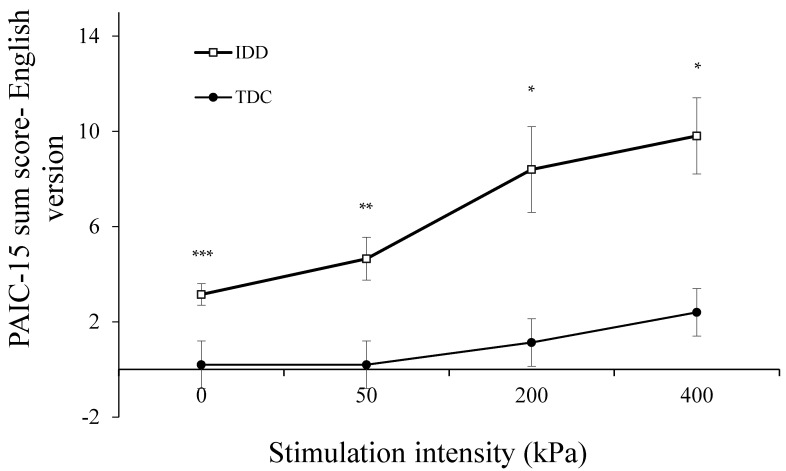
The sum PAIC-15 scores of the English version for individuals with IDD and TDCs. The PAIC-15 scores of the IDD group were significantly higher than those of TDCs in all the stimulation conditions (* *p* < 0.01; ** *p* < 0.001, *** *p* < 0.0001). Values denote the group mean ± SEM.

**Figure 2 brainsci-11-01254-f002:**
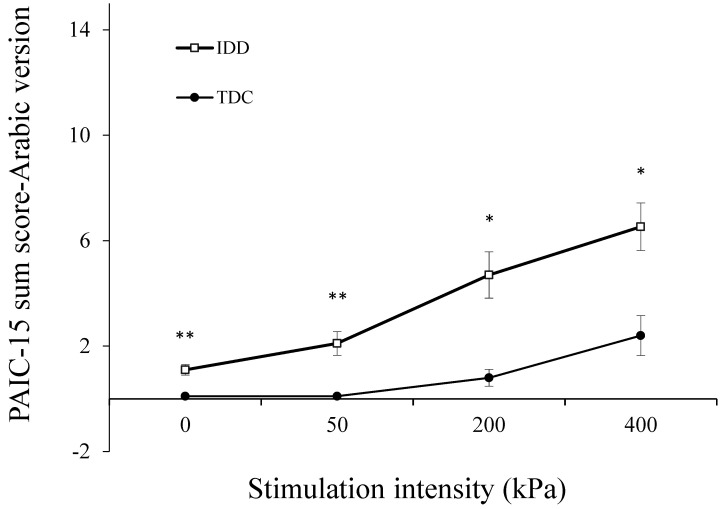
The sum PAIC-15 scores of the Arabic version for individuals with IDD and TDCs. The PAIC-15 scores of the IDD group were significantly higher than those of TDCs in all the stimulation conditions (* *p* < 0.05; ** *p* < 0.01). Values denote the group mean ± SEM.

**Figure 3 brainsci-11-01254-f003:**
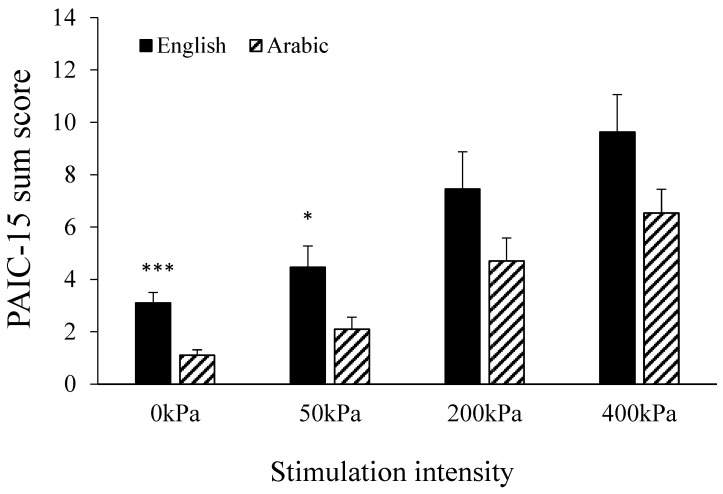
The sum PAIC-15 scores of the English version as compared to those of the Arabic version for the entire cohort. The scores at rest and during 50 kPa were higher using the English version (between groups: * *p* < 0.05, *** *p* < 0.001). However, the scores during the noxious stimuli (200 and 400 kPa) were similar for the two versions. Values denote the group mean ± SEM.

**Table 1 brainsci-11-01254-t001:** The study groups.

	Intellectual and Developmental Disability	Typically Developing Controls	*p*-Value ^#^
Number	30	15	
Females (*n*, %)	(53) 16	8 (53)	1
Age (M, SD)	35.5 (6)	31.3 (8)	0.07
IDD etiology (*n*, %):			
Cerebral palsy	13 (43)	--	
Down syndrome	8 (27)	--	
Unspecified IDD	9 (30)	--	
IDD level (*n*, %):			
Mild	16 (53)	--	
Mild–moderate	7 (23)	--	
Moderate	7 (23)	--	

^#^ = Mann–Whitney test or t-test between groups.

**Table 2 brainsci-11-01254-t002:** Frequency of occurrence of each PAIC-15 item during 400 kPa stimulation: comparison between groups and between the English and Arabic versions.

	English			Arabic			ICC
	IDD	TDCs	Total	IDD	TDCs	Total	
Frowning	9 (30)	4 (27)	13	9 (30)	4 (27)	13	1.00
Narrowing eyes	17 (57)	5 (33)	22	14 (47)	3 (20)	17	0.86
Raising upper lip	18 (60) **	2 (13)	20	5 (17)	1 (7)	6	0.49 ^^
Opening mouth	22 (73) ***	2 (13)	24	17 (57) **	1 (7)	18	0.86
Looking tense	17 (57) **	1 (7)	18	17 (57) *	3 (20)	20	0.39
Freezing	18 (60) *	4 (27)	22	14 (47)	5 (33)	19	0.58
Guarding	3 (10)	0	3	5 (17)	0	5	0.65
Resisting care	2 (7)	0	2	1 (3)	0	1	0.70
Rubbing	2 (7)	0	2	4 (13)	0	4	0.56
Restlessness	2 (7)	0	2	2 (7)	0	2	0.42
Pain-related words	9 (30)	1 (7)	10	6 (20)	0	6	0.89
Shouting	4 (13)	0	4	0	0	0	--- ^
Groaning	5 (17)	0	5	5 (17)	0	5	0.75
Mumbling	3 (10)	1 (7)	4	0	0	0	--- ^
Complaining	4 (13)	0	4	6 (20)	0	6	0.47
SUM score (m ± SD)	9.8(7) *	2.4(3)	6.1(4)	6.5(4) *	2.4(2)	4.4(3)	0.89

ICC = interclass correlation coefficients between the languages. For each PAIC item, the numbers in parentheses are % out of the individuals in each group and the asterisks signify the results of χ^2^ tests comparing the frequencies between the IDD and TDCs within each language (* *p* < 0.05; ** *p* < 0.01, *** *p* < 0.001). ^ are the results of χ^2^ tests comparing the total frequency (IDD and TDC) between languages (^ *p* < 0.05; ^^ *p* < 0.01). For the sum score, the numbers in parentheses are standard deviations and the asterisks signify the results of t-test between the IDD and TDCs within each language (* *p* < 0.05). The third column in each language is the average of the two groups.

## Data Availability

The data presented in this study are available on request from the corresponding author pending institutional approval.

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
