# Peer review of "Pain Behavior of People with Intellectual and Developmental Disabilities Coded with the New PAIC-15 and Validation of Its Arabic Translation"

_brainsci, 2021, doi:10.3390/brainsci11101254_

Round 1

Reviewer 1 Report

This is a very interesting study, in which the use of PAIC15 for the observation of pain behaviour in people with intellectual and developmental disabilities (IDD) is studied.

The study is well performed, and written down nicely. However, the wording of the aims needs to be more precise. Aims are presented a little different in abstract, introduction and discussion. Please think carefully about use of terms.

Usability suggests a more broader concept than what is studied here. Lines 345-350 mention ‘applicability’ and ‘feasibility’ of PAIC15, and should not be included in the first paragraph of the discussion. This is new (indirect) information, and to my opinion not the main result of this study. Also, I would suggest to replace or delete in line 445 ‘is user-friendly, does not require special professional training’ as this is not concluded from results of this study.

Does the word ‘validation’ mean that validity is studied, in COSMIN terminology (e.g. in discussion line 397 ‘criterion validity’)? Did the authors have hypotheses on scores on forehand, that perhaps can be included in the introduction or methods section? This would help the reader, and point to the most relevant comparisons. In this light, lines 395 onwards helped me.

Given the fact that scores were different between groups IDD and TDC, I hesitate whether it is appropriate to present analyses on both groups together (e.g. lines 302-306, 320-303, figure 3). The other results are already interesting and enough to realize the aims?  

Please add more information about the coders in the methods section. What was the background of the coders, e.g. language and experience/ profession? How many coders were there? It is very interesting that there were differences in scoring for some items between the languages (lines 409 onwards). Could this a result of differences in coders or culture?

Minor points

  • Please be precise in the use of the name of the PAIC15. To my knowledge it is ‘Pain Assessment in Impaired Cognition (PAIC15) scale’, so leave out ‘individuals with’.
  • Title page, asterix is used for two purposes (correspondence and equal contribution), therefore unclear.
  • Abstract line 20: instead of ‘behavioral tool’ I would call it an ‘observational tool’, the same for line 26 ‘behavioral scores’ change to ‘observational scores’.
  • Methods: given the chosen tests, were PAIC15-scores normally distributed?
  • Section 5 line 224: this is not ‘conclusions’ but ‘results’
  • Figures: I would prefer whiskers of 95% confidence intervals instead of SEM
  • Table 2, please add ‘400 kPA’ in title
  • Delete lines 337-338
  • Mention use of video recordings as a strong point in discussion?

Reviewer 2 Report

This manuscript reports about observation of pain-related behaviour in response to pressure stimuli in persons with intellectual and development disabilities (IDD) and controls recruited from the same area, to validate the PAIC15 pain scale. The design involves comparison of ratings of pain-related behaviour as phrased in the original English version with an Arabic translation. Such study is uncommon in the literature. Interesting decisions about the Arabic translation have been described in detail as are the statistical analyses, with methods justified. Overall, the work impresses as accurate and well-written, with sufficient detail to understand the design and probably also to replicate it as needed. The discussion offers possible explanations of findings and nicely provides recommendations for further translation testing and further research.

Minor

Perhaps avoid referring to recent development for a project that ended 4 years ago (line 55)

The atypical behaviors frequently reported among persons with IDD (line 65) needs a reference. I can see references only in the Discussion  (line 384-386).

Could you justify the unequal numbers in the two groups as this usually involves inefficiency for power

How did you manage missing values due to “non scorable” (line 348)? Please add to the Methods.

Table 2: I do not understand the footnote indicated with a Ù in the ICC column referring to Chi-square tests

Table 1: (appended), numbers under 20 definitely need no decimal percentage which suggests over precision (such as 53.3% for 8 of 15)

I do not understand what is “the single PAIC-15 item”, I cannot see this item defined before (line 220)

In the text of the Results, you may consider using the exact p-values, as the cut offs are not mutually exclusive and reporting exact values whether or not significant represents good practice. Further, you may use consistently more decimal spaces for p=0.1 (line 311) such as in line 304 (p=0.13).

The sentence about Cronbach’s alpha “in the present sample” is similar in para 5.3 and 5.5 (lines 267 and 299) but is better distinguished by referring to the precise sample.

Phrasing of “at rest” is confusing (line 306 and other places), does this mean no stimulus rather than no movement?

A section about patents (336-338) seems off.

The discussion may add consideration of:

-the development of the PAIC15 scale for those with impaired cognition, and any differences between groups such as dementia and IDD (showing many atypical, individual behaviors) and whether a single scale for apparently very different groups is justified. Would a pain scale for IDD benefit from description of specific behavior for the individual (e.g., which verbalizations)?

-awareness of social rules and monitoring by others whether this also plays a role in typically developing people inhibiting their responses, also awareness of their behavior being video taped (lines 359-370). Please also report research about infants, if available, and add age of the children in line 359.

-lower scores with the Arabic version are presented as possibly coincidental (line 422), which is not fully satisfying.

Reference list: inconsistent use of et al. (ref 43 to drop 2 names for example, but there are many references with many more than 5 authors)

There are a few typos, most notably in the name of the last author.
